# Behaviour change support education in chronic disease: An international focus group study with undergraduate students and academic educators in nursing, pharmacy and sport science disciplines

Isa Brito Félix[1]*, Carla Nascimento[2], Patrícia Pereira[2], Katja Braam[3], Cathal Cadogan[4], Judith Strawbridge[4], Leona Cilar Budler[5], Lucija Gosak[5], Nuno Pimenta[6,7,8], Mara Pereira Guerreiro[9]

1 Egas Moniz Center for Interdisciplinary Research (CiiEM), Egas Moniz School of Health & Science, Caparica, Almada, Portugal, 2 Nursing Research, Innovation and Development Centre of Lisbon (CIDNUR), School of Nursing, Universidade de Lisboa, Lisbon, Portugal, 3 Faculty of Health, Sports and Welfare, Inholland University of Applied Sciences, Haarlem, The Netherlands, 4 School of Pharmacy and Biomolecular Sciences, Royal College of Surgeons in Ireland, Dublin, Ireland, 5 Faculty of Health Sciences, University of Maribor, Maribor, Slovenia, 6 Sport Sciences School of Rio Maior, Polytechnic Institute of Santarém, Santarém, Portugal, 7 Interdisciplinary Centre for the Study of Human Performance, Faculty of Human Kinetics, Cruz-Quebrada, Portugal, 8 SPRINT – Sport Physical activity and health Research and INnovation CenTer, Santarém Polytechnic University, Portugal 9 Egas Moniz Center for Interdisciplinary Research (CiiEM), Egas Moniz School of Health & Science, Caparica, Almada, Portugal,

* ifelix@egasmoniz.edu.pt

## Abstract

### Introduction

Supporting behaviour change for the self-management of chronic diseases is a key competency for health professionals. Little is known about how undergraduate students and academic educators across disciplines and European contexts perceive current education, identify unmet needs relative to a European competency frame-work, and view the implementation of novel educational tools. To address this gap, we explored students' and educators' perspectives, focusing on unmet needs, views on interprofessional education, preferred modalities for using a MOOC, case studies and a simulation software, and perceived facilitators and barriers to use.

### Methods

Twelve online focus groups were conducted separately with a purposive sample of 39 undergraduate students (seven groups) and 27 academic educators (5 groups). Recruitment spanned national and transnational levels across seven European countries, and included participants from nursing, pharmacy and sport science disciplines within the focus groups. The discussions were recorded, transcribed verbatim, and analysed using thematic analysis.

**Data availability statement:** The datasets generated and/or analysed during the current study cannot be made openly available because participants did not provide consent for open sharing at the time of this study. The data are currently pseudonymised. Requests for further information regarding data access should be directed to the Ethics Committee, Instituto Politécnico de Santarém (IPSantarém), at comissaodeetica@ipsantarem.pt.

**Funding:** This project has received funding from the Erasmus+ Programme of the European Union under the grant agreement no. 2019–1-PT01-KA203–061389. The Funder had no role in the design of the study and collection, analysis, and interpretation of data and in writing the manuscript. The European Commission's support for the production of this publication does not constitute an endorsement of the contents, which reflect the views only of the authors, and the Commission cannot be held responsible for any use which may be made of the information contained therein.

**Competing interests:** The authors have declared that no competing interests exist.

## Results

Unmet needs in behaviour change support education were identified (e.g., models and theories, behaviour change techniques, work in partnership to prioritise target behaviours), detailing contributing factors and strategies for their development. Another theme was interprofessional behaviour change education, highlighting perceived benefits, barriers, such as limited training of educators, and implementation strategies. The third theme, on educational products, yielded insights on barriers and facilitators of use, as well as implementation within undergraduate programme.

## Conclusion

This study highlights the unmet needs in behaviour change support education, the potential of interprofessional education (IPE) and innovative educational products to address these gaps. Findings suggest the need for integrating behaviour change support education into undergraduate curricula, enhancing interprofessional learning on this topic and leveraging digital tools to better equip future professionals in chronic disease management. Strategies emerging from the data can guide these endeavours.

## Background

Noncommunicable diseases (NCDs), also known as chronic diseases, are the predominant cause of morbidity and mortality worldwide. In 2019, noncommunicable diseases were responsible for 74% of deaths globally [1]. These diseases represent a significant burden for the persons affected, their families and the health systems [2].

Managing chronic disease is a major challenge for health systems and the workforce around the globe. Healthcare and other professionals are expected to support behaviour change for the self-management of chronic disease, including, for instance, interventions to improve diet or increase physical activity. Studies show that many healthcare professionals frequently miss the opportunity to provide behaviour change advice, even when it is perceived as needed [3,4]. An online survey (n = 1338) showed that only 40% of the nurses, midwives and healthcare support workers were comfortable having conversations around behaviour such as alcohol intake, physical activity, healthy weight, and diet [5].

A systematic review synthesised barriers and facilitators for delivering behaviour change interventions by healthcare professionals [6].Two key barriers identified were the perception of insufficient skills and a lack of confidence in supporting behaviour change. Additionally, the review highlighted the absence of adequate behaviour change training as a common barrier across various healthcare professions, including pharmacists, midwives, nurses, and general practitioners [6].

Despite these challenges, limited research exists on education relating to behaviour change support for undergraduate students, and the available evidence suggests that there are opportunities for improvement in current curricula. For instance, Van Hooft and colleagues [7] found that behaviour change education for

Dutch nursing students was largely confined to theoretical models and communication skills. The existing evidence highlights the need to address educational gaps and better equip professionals to support behaviour change in self-management of chronic disease.

Despite growing recognition that behaviour change support is a core professional competency, undergraduate curricula often provide limited, inconsistent, and largely theoretical preparation, with variable opportunities for interprofessional and practice-oriented learning. Moreover, little is known about how undergraduate students and academic educators across disciplines and European contexts perceive current provision, what they identify as unmet needs, and what conditions would support the use of emerging educational tools designed to address these gaps. Developing educational products does not in itself ensure that content and delivery are relevant, or their adoption in practice. More recently, a scoping review of 49 studies on new digital methods for teaching practical skills has reinforced this point, emphasising the need to engage students and teachers during development [8].

To respond to these challenges, the Erasmus+ Train4Health project (2019–2022, https://www.train4health.eu) developed a European interprofessional competency framework [9], a linked transnational curriculum [10] and a set of educational products (MOOC, case studies, simulation web application). The present study explored students' and educators' perspectives on these products focusing on unmet needs relative to the competency framework, views on interprofessional education, preferred modalities for using the products, and perceived facilitators and barriers to use.

In addressing these challenges, this study seeks to provide a transferable approach to guide curricular innovation in diverse contexts by demonstrating transnational user engagement to guide design, development and implementation decisions, and by showcasing a framework-based approach to needs assessment.

## Methods

A qualitative study was conducted through online focus groups involving students and academic educators, via videoconferencing.

Focus groups were chosen to support discussion between students and educators across disciplines and countries, enabling participants to debate views and collectively explore opinions; this interaction can generate insights that would not emerge as readily in individual interviews.

The research team was comprised by a study leader (IBF), who acted as site lead for Portuguese consortium partners, plus site leads for consortium partners in Ireland, the Netherlands and Slovenia (CC, KB, LG), and supporting researchers in these study sites (CN, PP, NP and MPG in Portugal; JS in Ireland and LCB in Slovenia). Site leads and supporting researchers who moderated focus groups held a PhD (CC, CN, KB, MPG, PP) or an MSc (IBF, LG) and had experience relevant to conducting the focus group study. The research team brought professional backgrounds in pharmacy, nursing and sport science; eight researchers were female and two were male (site leads: three female, one male; supporting researchers: five female, one male). Some moderators had pre-existing professional relationships with academic educator participants within their own institutions, and some student participants may have recognised moderators as educators. All moderators were members of the funded consortium underpinning this work; this positioning may have shaped facilitation and interpretation. To mitigate this, participants were informed of the study aims and researchers' roles during consent, topic guides were collaboratively developed and piloted, moderation procedures were standardised, and analysis was undertaken by more than one coder with consensus meetings to agree coding decisions and themes. Furthermore, the study leader maintained regular contact with site leads throughout the study.

Ethical approval was granted from Sport Science School of Rio Maior, Portugal (reference number: 092020Desporto). This study is reported in accordance with the COREQ checklist [11].

### Participant selection and recruitment

Purposive sampling of students and academic educators was used, at a national level (consortium partners) and transnational level (other higher education institutions and international organisations, such as the International Pharmaceutical

Students' Federation, European Pharmaceutical Students' Association, European Federation for Educators in Nursing Science and European Association of Faculties of Pharmacy).

To ensure informed opinions, students had to be in their final year of their nursing, pharmacy, or sport science degree or having a current, or past role in students' unions or higher education institutions' boards. Student leadership roles were considered an alternative to final-year status because these students typically have exposure to quality processes and cross-cohort student needs, and are often involved in discussions on educational provision. The criterion for recruiting academic educators was having experience or interest in behaviour change support education in nursing, pharmacy, or sport science.

A total of 12 online focus groups were conducted, seven with students and five with academic educators from pharmacy, nursing and sport science. Each focus group lasted, on average, approximately 90 minutes (range 29–120 min). Size ranged from four to seven participants (mean 5.5 participants). Fig 1 illustrates the distribution of the focus groups.

Site leads were responsible to recruit participants by email in their respective Institutions (Table 1), aided by an information leaflet and a consent form. All participants provided written informed consent prior to data collection.

Participants received a certificate of participation; no financial incentive was offered.

Recruitment took place between 20 July and 15 September 2020.

## Data collection

Separate topic guides were developed for students and academic educators, to capture the perspectives of the two groups on the same issues, informed by the literature and the study objectives.

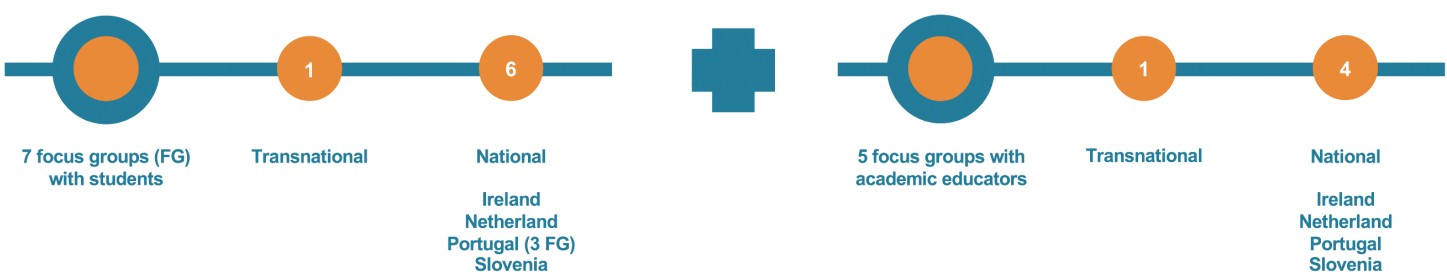

**Fig 1. Overview of conducted focus groups.**

**Table 1. Higher education institutions where participants were recruited.**

| Country | Institution |
|---|---|
| Portugal (consortium partner) | Nursing School of Lisbon (ESEL) |
| | Sport Science School of Rio Maior (ESDRM) |
| | Faculty of Pharmacy, University of Lisbon (FFUL) |
| Ireland (consortium partner) | Royal College of Surgeons in Ireland (RCSI) |
| Lithuania | Vilnius University |
| | Lithuanian Sports University |
| Malta | University of Malta |
| Norway | Western Norway University of Applied Sciences |
| Slovenia (consortium partner) | University of Maribor (UM) |
| the Netherlands (consortium partner) | Inholland University of Applied Sciences |

An English version of these data collection instruments was developed in close collaboration with all site leads, to ensure applicability across countries. Both topic guides were translated from English to Portuguese, and piloted in separate online focus groups with students (n=4) and academic educators (n=6). Focus groups recordings were discussed within the Portuguese research team; subsequently topic guides in Portuguese and English were refined which mainly involved reducing the length of the topic guide. Then, questions and prompts were translated from English into Dutch and Slovenian by the respective site leads.

Data collection was also aided by a brief PowerPoint presentation, shared during focus group sessions, to set the context about the European competency framework [9] and describe the products. Only the Slovenian site translated this document.

Focus groups at a national level were conducted in each country's respective language (Portuguese, Slovenian, Dutch and English), to facilitate communication, whilst transnational focus groups were conducted in English.

Each focus group was run by a moderator, in charge of fostering an active and open discussion, and an assistant researcher, tasked with taking notes. Moderators were usually the site leads; however, the transnational focus groups were moderated by MPG, based in Portugal.

Recordings were obtained through Microsoft Teams or Zoom platforms, depending on institutional adoption and availability. Procedures were standardised across platforms.

Field work took place between September and November 2020.

## Data management and analysis

All focus groups recordings were transcribed *verbatim* in the language in which they were conducted. A consistent format was used to layout the transcripts. Each participant was assigned a unique identifier, starting with the initials of the country (PT, IR, SL, NL, EU for respectively Portugal, Ireland, Slovenia, the Netherlands and countries), category (AE or S, for academic educator and student), degree (N, P, SS, for, respectively, nursing, pharmacy or sport science) and, finally, a sequential number within each category.

Data were analysed based on Braun & Clarke [12] (thematic analysis), applying a combined deductive-inductive approach. An initial coding framework was developed deductively from the study aims and topic guide and then refined inductively using codes emerging from pilot focus groups data in the Portuguese site. The resulting coding framework was provided to all site leads as an excel template with instructions for coding, the definition of each code, and an illustration of textual data indexed to the code.

This initial codebook was used for data analysis in each site, performed preferably by two researchers in the language in which the focus group was conducted, including at least the focus group moderator. Each site-level codebook was refined inductively through analysis of datasets across sites, and then merged into a common codebook across sites in English, which was consolidated iteratively.

The complete data file was organised around concepts to identify patterns and linkages between them and to finally be able to answer the study aims. A consensus meeting between each study site and the study leader (IBF) was held to discuss uncertainties and to reach consensus on codes, themes and sub-themes.

## Results

### Participants characteristics

In total, 39 students and 27 academic educators took part in the focus groups. Participants were from the following institutions: Nursing School of Lisbon, Royal College of Surgeons in Ireland, Sport Sciences School of Rio Maior, Faculty of Pharmacy, University of Lisbon, University of Maribor, InHolland University of Applied Sciences, Polytechnic of Leiria, Egas Moniz School of Health & Science, University of Porto, Utrecht University, Lithuania Sport University, University of Malta, University of Coimbra, Vilnius University and Western Norway University of Applied Sciences. Thirteen participants did not report their university affiliation.

The average age of the students was 22 years, 56.4% were female, 11 were nursing students, 12 sport science students and 16 pharmacy students. Age and number of years as educator data from one participant were missing and therefore were excluded from analysis. The academic educators had an average of age of 50 years (min. 39; max.61). The majority were females (74%), 14 (51.9%) had a Master's degree and 13 (48.1%) had a Doctoral degree. Nine academic educators were from institutions outside the consortium.

## Findings

The themes and sub-themes were developed from the analysis are illustrated in Fig 2. Regarding the sub-themes under the educational products, each one observed consistently across all of them, except for facilitators, which emerged only for the MOOC and simulation software.

## Unmet needs in behaviour change support education

Participants described a mismatch between the behaviour change support competencies expected of future professionals and the training provided in undergraduate curricula. These perceived gaps were attributed to limited opportunities for practical application and weak integration of behaviour change content across curricular modules, leading participants to propose a range of strategies to strengthen competency development and improving curricular alignment.

This theme comprises three interrelated sub-themes: (i) top unmet competencies, identifying the competency areas most frequently perceived as challenging; (ii) contributing factors, explaining why these gaps persist; and (iii) strategies to develop these competencies, outlining practical suggestions for addressing the identified needs.

First, participants discussed difficulties and gaps in behaviour change support education in undergraduate curricula in relation to the European competency framework. Specifically, difficulties were voiced in achieving five competencies (BC6, 8, 9, 12 and 13), illustrated in Fig 3.

When talking about knowledge on behaviour change theories and models, a student said "*I think we don't deal with it very much. So, it is a difficult one (…).*" [PTSP18], illustrating how perceived difficulties in achieving desired competencies as future professionals may be linked with unmet needs in undergraduate curricula. Moreover, four students perceived a

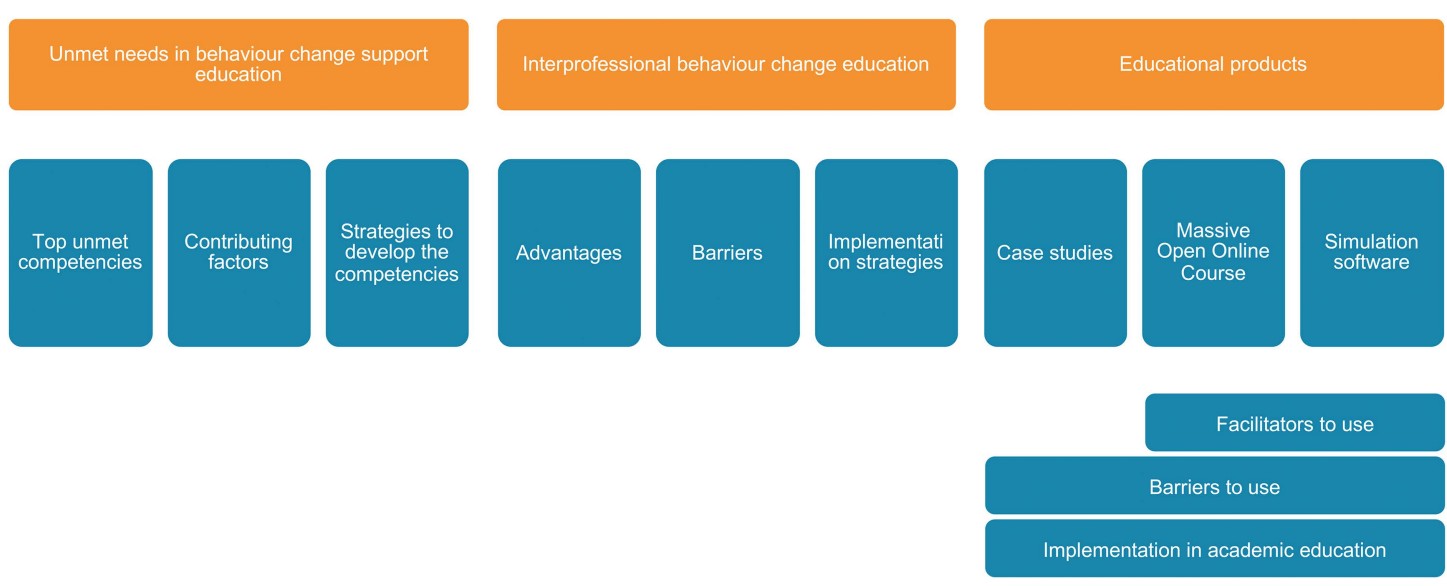

**Fig 2. Overview of the themes and sub-themes identified in the analysis.**

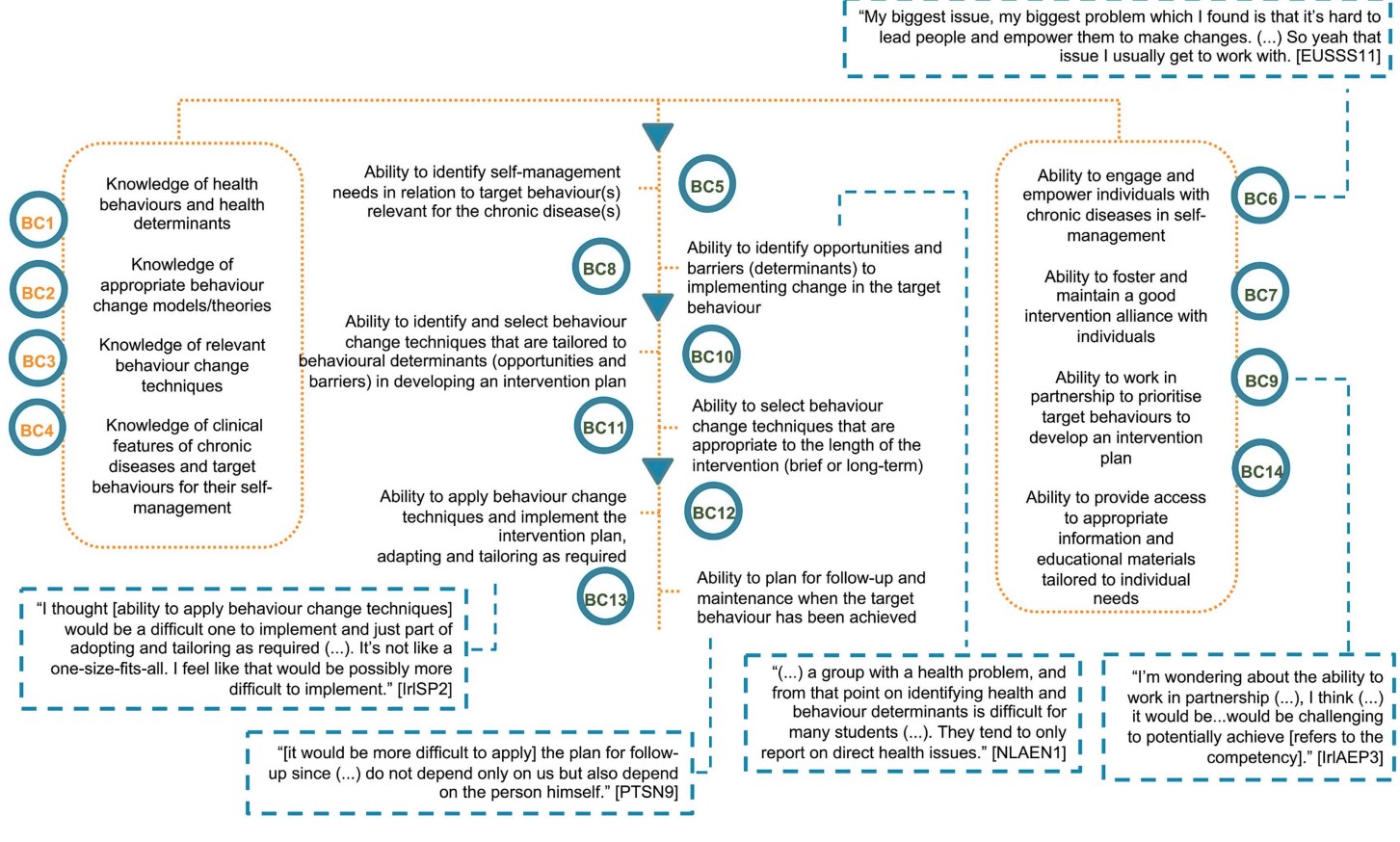

**Fig 3. Perceived difficulties in achieving behaviour change competencies as professionals.**

gap in content about behaviour change techniques (BCTs). As one student put it: "*There is little knowledge of behaviour change techniques… (…) it may be a gap that I (…) makes me less comfortable (...)*" [PTSP10]. This view did not receive unequivocal endorsement, as one academic educator from another country mentioned addressing BCTs, although perhaps not in great length: "*We definitely do touch on the behaviour change techniques*" [IrlAEP1].

Only students discussed concerns about competencies on models, theories and behaviour change techniques (BC2 and 12, respectively, Fig 3), while concerns about the ability to work in partnership to prioritise target behaviours (BC9) were only expressed by academic educators [IrlAEP3; EUAEN7].

When exploring reasons underpinning difficulties in achieving competencies encompassed in the competency framework (Fig 3), factors emerged beyond topics encompassed in curricula. Eleven students and two academic educators agreed on the lack of a practical approach during undergraduate education, discernible across nursing, pharmacy and sport science.

As one student said: "*The education should focus more on BCT's when providing practical training*" [NLSSS1]. Five students and one academic educator recognised the absence of an integrated approach across subjects in degrees as a reason underpinning difficulties in achieving the desirable competencies:

"*We definitely do touch on the behaviour change techniques, stages of change model, but they are generally kind of isolated sessions, health promotion sessions or something like that. There isn't a whole lot of active integration into general clinical teaching, therapeutics teaching*" [IrlAEP1].

Talking about this issue, one of the five students contended that lack of integration also existed across institutions and countries, even for the same degree:

"*I think, at least for pharmacy… students here [in France] they might have a lot of variation on this [refers to behaviour change support education]. I think in some countries maybe there is a lot of attention to the subject and not in others*" [EUSP8].

In addition, shortage of time was identified as an issue influencing how students are equipped to demonstrate behaviour change support competencies in the future. Three academic educators and three students associated time shortage with rigidity in curricula, which jeopardised changes in content or the implementation of new approaches: *"(...) It's not easy to... to add more content or topics"* [NLAESS1].

Participants proposed strategies to improve education, mainly focused on instructional methods, such as role-play (n = 6), case-based learning (n = 7), problem-based learning (n = 2) and training with expert patients (n = 10), as illustrated by participants' accounts:

"*Actually acting it out with somebody and coming up with scenarios would be the best way to get the message across*" [IrlSP5].

"*The best way is having several simulations with colleagues or … and especially with real cases and persons*" [PTAESS1].

The latter was only suggested by students. A minority of academic educators suggested an overarching strategy of step-wise complexity - "*I think starting with (...) a standard situation and then adding complexity in different ways*" [EUAEN7]; no participant was directly opposing this strategy.

### Interprofessional behaviour change education

Participants viewed interprofessional behaviour change education as valuable for preparing students to deliver coordinated, patient-centred behaviour change support. They emphasised that learning alongside other disciplines can clarify professional roles and strengthen future collaboration. At the same time, participants highlighted that uptake remains uneven, largely due to structural and pedagogical constraints, leading them to propose pragmatic approaches to implementation. Overall, this theme captures a tension between perceived benefits and practical constraints within curricula and institutions. It comprises three interrelated sub-themes: (i) perceived advantages of interprofessional behaviour change education; (ii) barriers to its delivery; and (iii) implementation strategies within undergraduate programmes.

In terms of advantages, there was unanimous agreement on the value of interprofessional behaviour change education. Central to these accounts was a common view that interprofessional behaviour change education enables synergistic collaboration between different professionals. Six participants – four students and two academic educators – argued that it enables students to foster their own role: *"(...) knowing where your responsibilities as a sport science professional end and others start is needed"* [NLSSS1]. Moreover, both students and academic educators (n = 10) stated that it facilitates envisaging future teamwork, as exemplified by the quote below:

"*I think that when you actually work together and you see the other perspective and the process from the other side that is actually the start of better interprofessional collaboration in the future*" [EUSP8].

 

There was a consistent view among academic educators and students who agreed that Interprofessional Education (IPE) may enhance the knowledge of each other's professional role: *"it is very important for all of us to know the role of each of the different professionals, and to try, let's say, to fill each area with each one"* [PTSN6].

Furthermore, three students revealed that the interprofessional behaviour change education contributes to a concerted action among healthcare professionals. As one student explained:

> *"(…) From sport science, nurse point of view, pharmacy… and they can give to the patient a better plan on how to solve his problem faster. He doesn't need to go more than one time, or two times to every specialist. So, if more specialists join together on the same problem, on the same chronic disease it could help"* [EUSSS10].

Other advantages of the interprofessional education highlighted by the participants included gaining knowledge for practice, a point mentioned by only one student - *"Yes (...)* [the IPE is important] *for the acquisition of knowledge for practice"* [PTSN9]. Additionally, the benefit to patients was emphasised by both groups of participants in Portugal: *"There is clearly a benefit, this interdisciplinary idea is an idea that we all defend (...) so if there is a gain is for the patient"* [PTAEN2].

However, participants also pointed to barriers that help explain why involvement in interprofessional behaviour change education remains limited. With few exceptions, participants described engagement as incipient and highlighted six key barriers, often reflecting challenges in IPE more broadly.

One of the most frequent barriers raised by academic educators from different countries (n = 5), was the perceived limitation in their own training to provide IPE, as illustrated by this quote: *"(...) so academic educators' training is also an issue that I anticipate"* [PTAEN3]. This lack of sufficient training ties directly to the need for an interprofessional team to support implementation efforts (n = 2), which was seen *"as difficult to achieve, but again could be achieved by upskilling faculty and co-teaching perhaps"* [IrlAEP3]. Additionally, differences in educational pathways across disciplines, such as pharmacy, nursing, and sport science, was perceived as another barrier to IPE (n = 3). These disciplines follow distinct educational pathways, leading to different stages of student development which, from the participants' standpoint, makes it difficult to adopt interprofessional approaches. As one student explained, *"(...) this [interprofessional education] also brings some difficulties, because it is necessary for students to be on a similar level of training"* [PTSP18]. Logistical issues were also raised. For example, one academic educator commented on the complexity of coordinating schedules across different academic units:

> *"(…) the challenge is how to get the different academic units from different specialties to agree on a time, on a schedule, on a timetable, where the interprofessional education can be held"* [EUAEP1].

Insufficiencies in infrastructure and equipment to support IPE was mentioned by others, as illustrated by of narrative of this academic educator:

> *" (...) we should have facilities where students can work together. And then, having group work, and here's the big question and I now talk about the facilities I know, where I work, access to a computer would be interesting"* [PTAEP5].

Finally, low prioritisation of IPE within the educational setting was also seen as a barrier, with two divergent narratives emerging, with the first quote revealing a tension between the value placed on discipline-specific education and IPE:

> *"(...) it's nice to combine knowledge, but in the end, it's about having the skills for your own profession"* [NLSSS1].

> *"Education and learning the BCTs needs to be prioritised as well as working together... now this is not happening"* [NLSSE1].

Diverse perspectives were expressed to embed interprofessional behaviour change education, focusing on staffing models, timing within programmes and institutional arrangements. Most participants agreed on having academic educators from a range of disciplines during the course or modules. Talking about this issue, one academic educator said:

> "*I don't know whether a huge amount of upskilling for these particular competences would be required, particularly if you are co-teaching. I think the co-teaching would be brilliant. I think if you had a pharmacist and a psychologist or another professional and a patient, I think each could bring their own element to the table without a huge amount of upskilling*" [IrlAEP1].

Concerning the stage of the learning journey where IPE is implemented, some felt that it should be implemented in early on - "maybe a multidisciplinary task can help in introducing to the other disciplines in an earlier stage of the course" [NLSSS1] – with one student disagreeing "maybe it makes more sense at a time when we are about to go on an internship" [PTSP18].

Underlying this latter stance is the notion that the beginning of the learning journey should be devoted to gaining discipline-specific knowledge, reflecting once again the tension between the value placed on discipline-specific education and IPE. Notably, no academic educator offered their perspective on this matter.

Another suggestion that was put forward by both students and academic educators, was the implementation of common modules between different disciplines. One participant expressed this idea as follows: "(...) first of all, at the university level, they could implement some interprofessional activities or modules" [EUSP9]. This perspective was echoed by another participant, who highlighted that "(...) had a moment in our curriculum plan, where we could meet with students from other areas to discuss a case or use a simulation software, focusing on aspects requiring intervention by different professionals, it would be interesting" [PTSP18].

Building on the idea of interprofessional modules, another participant emphasised the importance of establishing institutional partnerships, sharing a successful experience to illustrate this point.

> "Actually, within (place), we have these groups of students from two courses and it's a really nice collaboration between our university and universities of applied science. So, I think for about 10 years now, we have had interprofessional courses where students work on cases together… but …you have to have a willingness… within all those institutions to bring students together. And it needs to grow. Start small and add the disciples every year" [EUAEN3].

### Educational products

Participants focused on what would enable or hinder uptake of the educational products in academic education, including practical constraints and the extent to which each product supports practice-oriented learning. Participants articulated facilitators only for the MOOC and the simulation software, whereas discussion of case studies focused predominantly on barriers and conditions for use. In analysing these data, we deductively considered the three products as subthemes (case studies, MOOC and simulation software) and then further derived other sub-themes within these – facilitators and barriers to their use and strategies for their implementation – as depicted in Fig 2.

### Facilitators and barriers of use the educational products

Convenience was highlighted by both students and educators as a facilitator for using the MOOC and simulation software. For instance, one participant noted: *"It's very convenient* [about the MOOC] *because you don't have to drive anywhere"* [SLSN3], while another added, *"It's probably much better logistically because it's online (...) you can be at home (...)"* [SLSN5]. Similarly, several students emphasised this idea in relation to the simulation software, mentioning its

*"accessibility and ease of use* (...) *that is always just a click away"* [PTSN6]. For the MOOC, **r**ecognition by professional regulators was also identified as another facilitating factor, as expressed by one student: *"if it* [the MOOC] *is accredited by professional regulators, it will certainly lead to greater adherence"* [PTSP18].

Regarding the simulation software, two students elaborated on the potential for learning through mistakes**:** *"I will learn from this error, I will do more tests…"* [PTSSS8]. Additionally, some students and educators highlighted its value for training for future practice: *"It is nice that the student can practice for future; not only a head full of new information, but also a true virtual reality patient to practice with"* [NLNE2].

The barriers to using the educational products were also pointed out by participants. Regarding the case studies, only students identified the following barriers:

- Rigidity of the curriculum: *"(...) the curricular plan as it is structured, nothing can be moved…I think it's the main barrier [to using the case studies]. It is important to find opportunities to apply the case studies"* [PTSN1].

- Time constraints: *"Time to cover all things..it's another main barrier [to using the case studies]"* [PTSN1].

- Educator-student ratio: *"the teacher-student ratio, where there is often a single teacher for 30 students or more and then integration or the possibility for students to use the case studies in a meaningful way becomes more difficult."* [PTSN4].

For the MOOC, the time constraints were a recurring concern, particularly in relation to fitting the course into existing curricula: "(...) I think 40 hours is quite a lot to squeeze into any curriculum" [IrlAEP5]. Additional barriers included lack of face-to-face interaction, organisation/educator's adherence and lack of students' motivation. The first barrier was expressed by students, with one sharing their experience: "*I'm the kind of person who learns more if I see something live than online..I can imagine that is a barrier for other colleagues as well*" [SLSN3]. Both students and educators recognised the potential challenge in achieving organisation/educator's adherence to novel teaching strategies, such as incorporating the MOOC into the current practice: *"(…) make use of these platforms (...) unconventional or more innovative, there may be a barrier (…)"* [PTSN6]. Finally, students expressed that the *"problem is the interest of students, because until someone forces you somewhere and you sit down, very few would opt for something like that. If this were not mandatory, very few students would probably choose it"* [SLSN5]. As described initially, the educational products will be available in English. Participants from countries where the mother tongue is different, emphasised the foreign language as a barrier to use the MOOC. For instance, a student mentioned that *"studying in English provides extra barriers to participate"* [NLSP2].

Concerning the simulation software, four main barriers to its use emerged. One of these, the foreign language**,** was also identified as a barrier for the MOOC. Additionally, the lack of digital literacy of the academic educators**,** academic educators' motivation and incompatibility with some computers were mentioned as specific barriers.

Two participants argued that academic educators might struggle with using the simulation software due to limited technological skills: *"Maybe a problem for some professors. (…) maybe not everyone is as technologically aware (…)"* [SLSN3]. This view was echoed by a student, who stated that academic educators *"use for 30, 40 years the same face-to-face method (...) it's very hard to adapt (...) to a new teaching method"* [EUSSS10]. Consequently, this lack of technological competence can influence the educators' motivation: *"the motivation of a certain professor is also a problem. So how much is he willing to invest in it, how much time, effort and everything to use the software"* [SLSSS2]. Lately, only a student highlighted that "i*t might not be able to be supported by all computers or devices, and maybe that could be a limitatio*n" [PTSN4].

### Implementation of the educational products

Participants' suggestions for implementation centred on how the educational products could be embedded within existing degree structures. Across accounts, implementation was discussed as a set of complementary decisions such as curricular integration and placement within the programme.

Firstly, participants identified as relevant implementing these three products through an interprofessional approach**.** The comment below illustrates this view for the case studies:

*"But I still think that actually solving the case together in a group from different professions would be the ideal situation, because in this way students can really see the way of thinking and the way a problem is approached whereas in the proposed method you would kind of get the same but you would not really see the process, you would only see the outcomes of the process"* [EUSP8].

Other statements reinforce this perspective on the MOOC from both students and academic educators:

*"It would be interesting* if it were developed not only by *a teacher from the nursing area but also by a sport teacher, a pharmaceutical science teacher,* and other educators who could contribute… Each could share a bit of their knowledge *(...)"* [PTSN5].

However, only academic educators highlighted this approach when discussing the simulation software:

"*I think the idea of involving several professionals in the simulation is brilliant. It would be much more engaging to place the person at the centre of this interaction and have them navigate between different professionals"* [PTAEP4].

Secondly, participants suggested integrating case studies and simulation software into the classes. Most students and educators expressed agreement with this approach:

*"I think the case studies can easily be incorporated into lectures or change the style of certain lectures to be case-based learning"* [IrlSP4].

*"A mix would be the best: lessons and integration of this product [simulation software] in the lessons. I would like that for myself"* [NLSKS1].

Additionally, students noted that the MOOC could even replace some existing modules: *"I think that in the current academic education, the MOOC could be a single unit and replace the lecture"* [EUSSS10]. This suggestion, however, was not extended to the other educational products.

Thirdly, participants expressed differing opinions regarding the appropriate timing for implementation of these products within the degree programme. Only one student suggested that the MOOC should be introduced at an early stage: *"the online course is feasible from the first year, with different degrees of difficulty. This is for Nursing (…) it is the course that I have more or less notion of how we are preparing for the job market"* [PTSN16].

However, perspectives were divided between introducing the case studies and the simulation software in an early stage or a later stage of the degree. For instance, an academic educator, based on their experience with internships being concentrated towards the end of the degree, suggested that case studies should be implemented *"(...) during the internships, when they will be with real people and will have the situations. Finally, there, they will gain significant decision-making skills and develop behavioural changes when they are already in contact with people"* [PTAEN2].

In contrast, some students recommended introducing case studies earlier:

*"[Case studies] from the 2nd year on are good because they have a whole clinical history behind them, and, therefore, everything related to a client requires much more work (…). In the second year, we are much more capable"* [PTSN16].

Regarding the simulation software, most participants who mentioned the timing emphasised that it should be implemented in the later stages of the degree:

*"I think that simulation software should be in the later stages so it could be some kind of test to see how we can implement our new knowledge, which we gathered through case studies and the MOOC platform"* [EUSSS11].

Only two students indicated that the simulation software could be introduced earlier, with one suggesting: *"Very useful as skills practice material for year 2"* [NLNE1].

When asked about the sequence for implementing the educational products, some participants agreed on introducing the MOOC first, followed by the case studies, and finally the simulation software, as reflected in the following statements:

*"(...), if we could refer students to a dedicated MOOC on it, then let them bring what they've learnt into a role play scenario or a practice setting"* [IrlAEP1].

*"To me, perhaps in terms of the sequence of this, I would understand that the theoretical part is e-learning, then case studies and then simulation, such an order seems to me the most logical possible"* [SLAEP5].

However, some participants were unable to identify a definitive sequence for implementing the educational products. One participant emphasised the importance of flexibility, stating:

*"People could still have the flexibility to choose the way to go because people have different learning times. Some people will be more comfortable in having, for example, the theoretical approach through the MOOC and then moving on to the case studies and the simulation, while others might prefer to try themselves out with the case studies or simulation first before approaching it the other way. So, I don't have one fixed answer on this…I think if there could be flexibility, that would be nice"* [EUSP9].

A similar idea was shared by an educator, who noted: *"it can be used all throughout the process, so I would not separate it into the beginning, middle or final stages. I think it offers a lot of flexibility and the situations can be addressed in many throughout the entire learning curriculum of the students. I think it has flexibility for that"* [EUAEN7]. Nonetheless, both students and educators expressed that the MOOC and simulation software should be implemented prior to the practical internship.

Other strategies for implementing the educational products were also suggested. For the case studies, participants proposed incorporating role playing - *"Taking this case and turning it into a role play would have more effect in this sense"* [PTSN1] – and group work - *"When you're in smaller groups, you're basically kind of actively involved in solving a problem"* [SLSSS2]. Additionally, the idea of increasing complexity was highlighted: *"You can start a case very simple and add complexity along the way"* [NLSSE2].

Regarding the MOOC, students and educators suggested that it should be complemented by other educational strategies: *"It might be useful as an add-on. Matched with simulation software, for example"* [NLPE3].

Ultimately, the implementation of the simulation software should be accompanied by reflection periods following its use. Addressing this point, a student remarked: *"I just wanted to add something...using the simulation software and after discussing with the teacher or even in groups"* [PTAESS1].

## Discussion

### Key findings

This study provides insights into unmet needs and perspectives on behaviour change support education among undergraduate students and academic educators in nursing, pharmacy and sport science disciplines. The findings underline challenges and opportunities in the implementation of interprofessional education and innovative educational products.

Findings suggest that behaviour change support is often positioned as a fragmented or peripheral element within healthcare curricula, rather than being developed as an integrated and practice-oriented competency set. This may help

explain why participants simultaneously report educational gaps while also identifying opportunities for interprofessional education and for the use of innovative educational products.

Participants emphasised the gaps in behaviour change support education, particularly regarding practical applications of behaviour change theories and techniques in existing curricula. Although provision varied across institutions and countries, the challenges described were similar across disciplines, suggesting shared structural constraints rather than degree-specific gaps (e.g., nursing, pharmacy or sport science). This points to the need for a transferable core approach to behaviour change support education that can be adapted to local curricular contexts, consistent with prior research reporting limited and uneven behaviour change training across healthcare curricula [13,14].

Current NICE guidelines state high-quality healthcare professional training in evidence-based approaches is crucial to improve persons outcomes in health behaviour change intervention [15]. There is evidence from reviews that behaviour change support training for healthcare professionals has a positive effect in clients. A meta-analysis with six studies suggests that healthcare professional training has a significant effect on clients' health behaviour for up to 12 months. Despite the small but significant improvement in clients' health behaviours, this finding supports the relevance of training in this field [16].

The unanimous support for interprofessional behaviour change education underscores its perceived value in fostering collaborative skills and role clarity from the participants' perspective. The recognition of the benefits of interprofessional education (IPE) is also well-documented in the literature [17–19] and the effective interprofessional education plays a role to achieve best competencies [20]. However, participants identified several barriers that may explain why its integration into undergraduate curricula remains limited and progress is slow. One key barrier is the lack of training for educators in IPE, which hinders its implementation. A qualitative study by Berghout [21] explored the experiences of nurse educators (n = 9) regarding their preparedness to teach IPE. The study reveals that none of the educators had undergone formal training tailored to teaching interprofessional collaboration. Most educators displayed only a partial understanding of interprofessional collaborative practice, and a small number were familiar with resources designed to support the integration of interprofessional collaboration into their teaching curricula. Recognising this gap, the International Working Group for Interprofessional Educators Competencies, Assessment, and Training is striving to establish a global consensus aimed at supporting IPE facilitation. This includes developing a comprehensive framework encompassing core competencies for IPE educators, assessment tools to evaluate these competencies and specialised training programmes to foster them [22]. These efforts are expected to enhance educators' preparedness and ultimately promote broader implementation of interprofessional education in healthcare curricula.

The finding highlighted the importance of determining the appropriate timing to introduce IPE on behaviour change support in higher education programmes. When implemented early, IPE can facilitate the understanding of interdisciplinary roles and foster collaboration from the outset. Conversely, introducing it at a later stage may prove more relevant to real-world professional practice. However, there is no clear consensus on this matter, although the inclusion of interprofessional modules has been well-received by participants as a promising approach. Notably, views on timing were articulated primarily by students, while academic educators did not comment on when IPE should be introduced. This may reflect limited autonomy over curriculum sequencing and a greater focus on feasibility constraints, whereas students drew on their lived experience of the learning journey and placements when considering when IPE would be most useful.

These perspectives align with the literature, such as the scoping review by Grace [23], which categorises IPE models into extracurricular or partially integrated models and fully integrated models. Extracurricular or partially integrated models involve elective or mandatory activities added to discipline-specific curricula and are easier to implement due to minimal restructuring. In contrast, fully integrated models embed IPE throughout the curriculum in a structured and progressive manner, reinforcing its role as a core component of education and preparing students incrementally for collaborative practice.

The simulation software was praised for its innovative approach, providing a low-risk environment where students can learn from mistakes and prepare effectively for future practice scenarios. It is positioned as a potentially invaluable resource for behaviour change support education, given the documented impact of virtual simulations in healthcare education, including improved knowledge, skills, and affective competencies [24,25].

Among the barriers identified, the lack of digital literacy among educators emerged as a concern. Participants highlighted that limited technological proficiency could impede educators' ability to integrate the simulation software in teaching. This challenge is further compounded by the broader issue of encouraging educators to adopt innovative instructional methods.

Systematic reviews indicate medium-to-low levels of digital competence among higher education educators [26]. Addressing these gaps through targeted and context-specific training initiatives may facilitate the adoption of digital tools, including simulation software, and support the integration of innovative methodologies in health education.

Regarding implementation, participants suggested a phased approach to integrating educational products, starting with theoretical MOOCs, progressing to case studies, and culminating in simulation software for skill application. However, this approach did not achieve unanimous agreement, with some participants advocating for flexible integration that enables the customisation of learning paths to accommodate varying educational needs and institutional contexts.

These preferences may reflect the heterogeneity of educational systems, resource availability, and professional priorities across disciplines and countries. This diversity underscores the importance of moving away from a one-size-fits-all approach. Instead, adaptive strategies are required to optimise the implementation of the educational products, ensuring their relevance and effective implementation in diverse educational contexts.

## Strengths and limitations

A strength of this paper lies in its comprehensive exploration of unmet needs in interprofessional behaviour change support education, particularly in the context of chronic disease management – a growing global challenge. By analysing perspectives through the lens of participants from multiple higher education institutions across Europe, this transnational approach provides nuanced insights that contribute to a broader understanding of this increasingly important area. Additionally, the structured, stakeholder-focused, and competency-based approach can be easily adapted by others seeking to develop or refine educational products. The input from key stakeholders on the barriers and facilitators to products use, ensures practical relevance and helps mitigate potential implementation challenges and strengthen the utility of these findings for curriculum design. Furthermore, the alignment of the educational products with a competency framework [9] ensures adherence to established professional standards, thus enhancing their adaptability and transferability across different educational and institutional settings beyond the scope of this study.

Another strength is the rigorous qualitative methodology employed, with a trained research team across multiple sites. A defined protocol outlining detailed procedures for data collection and analysis, ensured consistency and reliability. A shared codebook and a framework with illustrative quotes facilitated cross-site analysis and comparison, while regular team meetings resolved analytical uncertainties. This iterative and collaborative process not only ensured methodological robustness but also provided insights into managing transnational qualitative research.

Another aspect that merits discussion is that, during analysis, we noted increasing repetition across focus groups, with later discussions reinforcing the established thematic structure rather than introducing new themes. The dataset comprised 12 focus groups, which exceeds the number often reported as sufficient to reach thematic saturation in focus group research, while recognising that our multi-country and multidisciplinary sample was not fully homogeneous [27].

Despite these strengths, managing transnational focus groups posed challenges and limitations. The use of virtual platforms may have impacted participant engagement and the depth of discussions. Although prior studies suggest that videoconferencing can yield comparable data to in-person methods, these potential constraints should be considered when interpreting our findings [28,29]. Furthermore, we adopted a pragmatic approach to platform use, with each

institution using its standard videoconferencing system, which ensured familiarity for participants. We were unable to determine whether platform choice influenced participation or data quality. However, the functionalities used were limited to core features, i.e., live videoconferencing interaction and videorecording, that do not vary across common platforms [30]. Additionally, the purposive sampling strategy focused on specific disciplines such as nursing, pharmacy and sport science. While this ensured targeted insights, it might not fully capture the broader variability in educational practices and needs across other disciplines. Participant self-selection bias could have influenced the findings, as those with a stronger interest or opinions about behaviour change education may have been more likely to participate. Finally, recruiting students in leadership positions may over-representing students who are more engaged in governance and institutionally more literate than the broader student body, which may have strengthened the needs identified.

## Conclusions

This study identified unmet needs in behaviour change support education and examined stakeholder perspectives on interprofessional education (IPE) and three innovative educational products as potential ways to address these needs. Participants reported gaps in practice-oriented training and described barriers, facilitators and implementation considerations affecting IPE and the proposed products within existing curricula. These findings support strengthening skills-based and more integrated behaviour change support education into curricula, including learning strategies such as role-play, case-based learning and simulation activities. recommends embedding behaviour change support more coherently across curricula, rather than relying on isolated sessions, and adopting feasible models of IPE (e.g., co-teaching and shared interprofessional modules).

For implementation of the educational products, the findings support integrating case studies and simulation activities into taught sessions and considering a learning sequence that progresses from foundational knowledge (e.g., online learning) towards applied practice (e.g., simulation). Implementation planning should explicitly account for constraints and enablers identified in this study, including time demands, language accessibility in non-English-speaking contexts, variability in digital readiness and educator motivation. Overall, these recommendations point towards flexible, context-sensitive implementation supported by educator development to better prepare future professionals for collaborative chronic disease management.

## Acknowledgments

The authors are grateful for the support of the larger Train4Health team to the recruitment of participants. The authors are indebted to all study participants.

## Author contributions

**Conceptualization:** Isa Brito Félix, Nuno Pimenta, Mara Pereira Guerreiro.

**Data curation:** Isa Brito Félix, Carla Nascimento, Katja Braam, Judith Strawbridge, Mara Pereira Guerreiro.

**Formal analysis:** Isa Brito Félix, Carla Nascimento, Patrícia Pereira, Katja Braam, Cathal Cadogan, Judith Strawbridge, Leona Cilar Budler, Lucija Gosak.

**Funding acquisition:** Mara Pereira Guerreiro.

**Investigation:** Isa Brito Félix, Carla Nascimento, Patrícia Pereira, Katja Braam, Cathal Cadogan, Judith Strawbridge, Leona Cilar Budler, Lucija Gosak, Mara Pereira Guerreiro.

**Methodology:** Isa Brito Félix, Katja Braam, Cathal Cadogan, Nuno Pimenta, Mara Pereira Guerreiro.

**Project administration:** Isa Brito Félix, Mara Pereira Guerreiro.

**Supervision:** Mara Pereira Guerreiro.

**Validation:** Isa Brito Félix, Carla Nascimento, Patrícia Pereira, Cathal Cadogan, Judith Strawbridge, Leona Cilar Budler, Lucija Gosak.

**Visualization:** Carla Nascimento, Katja Braam, Cathal Cadogan, Judith Strawbridge, Leona Cilar Budler, Lucija Gosak.

**Writing – original draft:** Isa Brito Félix, Carla Nascimento, Katja Braam, Cathal Cadogan, Mara Pereira Guerreiro.

**Writing – review & editing:** Isa Brito Félix, Carla Nascimento, Patrícia Pereira, Katja Braam, Cathal Cadogan, Judith Strawbridge, Leona Cilar Budler, Lucija Gosak, Nuno Pimenta, Mara Pereira Guerreiro.

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
