## [Decision Letter · Decision Letter 0]

25 Dec 2025

Dear Dr. Félix,

Thank you for submitting your manuscript to PLOS ONE. After careful consideration, we feel that it has merit but does not fully meet PLOS ONE’s publication criteria as it currently stands. Therefore, we invite you to submit a revised version of the manuscript that addresses the points raised during the review process.

We look forward to receiving your revised manuscript.

Kind regards,

Elif Ulutaş Deniz

Academic Editor

PLOS One

Journal Requirements:

“This project has received funding from the Erasmus+ Programme of the European Union under the grant agreement no. 2019–1-PT01-KA203–061389. The Funder had no role in the design of the study and collection, analysis, and interpretation of data and in writing the manuscript. The European Commission’s support for the production of this publication does not constitute an endorsement of the contents, which reflect the views only of the authors, and the Commission cannot be held responsible for any use which may be made of the information contained therein.”

“This project has received funding from the Erasmus+ Programme of the European Union under the grant agreement no. 2019–1-PT01-KA203–061389. The Funder had no role in the design of the study and collection, analysis, and interpretation of data and in writing the manuscript. The European Commission’s support for the production of this publication does not constitute an endorsement of the contents, which reflect the views only of the authors, and the Commission cannot be held responsible for any use which may be made of the information contained therein.”

Reviewers' comments:

Reviewer's Responses to Questions

**Comments to the Author**

1. Is the manuscript technically sound, and do the data support the conclusions?

Reviewer #1: Yes

Reviewer #2: Yes

2. Has the statistical analysis been performed appropriately and rigorously?

Reviewer #1: I Don't Know

Reviewer #2: Yes

3. Have the authors made all data underlying the findings in their manuscript fully available?

Reviewer #1: Yes

Reviewer #2: Yes

4. Is the manuscript presented in an intelligible fashion and written in standard English?

Reviewer #1: Yes

Reviewer #2: Yes

Reviewer #1: This study offers a valuable contribution to the field, particularly by emphasizing the important role of behavioral change input in delivering patient-centered advice—especially in cases that require interdisciplinary collaboration among various healthcare professionals.

Nevertheless, a thorough review of the manuscript’s formatting and structural presentation is necessary. Specific areas for improvement include:

-Abstract: The abstract should be revised to improve clarity, coherence, and adherence to standard academic conventions.

-Text justification: Consistent text alignment is essential and should comply with the journal’s formatting guidelines.

-Use of bold text: The application of bold formatting appears excessive or unnecessary in several instances. Limiting its use will help maintain an appropriate academic tone.

-Bullet points: The format and usage of bullet points should be reviewed and revised to align with formal academic standards.

In conclusion, this is a well-conceived and relevant study with meaningful implications. The authors are encouraged to refine the manuscript’s presentation to enhance its overall academic quality and readability.

Reviewer #2: The abstract

• Line 7: The identified research gap requires further refinement. While the existing literature addresses behavior change education and interprofessional learning, the claim that certain perspectives “remain unexplored” necessitates either modification or stronger substantiation.

• The methods section of the abstract lacks specific details concerning the structure of the focus groups, such as whether they were discipline-specific or interprofessional, and whether students and teachers were combined or separated.

• Line 16: Grammatical issues require attention, such as the phrase “Participants were recruited at a national and transational level,” which should be corrected to “transnational.”

• Line 16: “The focus group discussions was recorded” should be “were recorded.”

• “This study highlights the unmet needs in behaviour change support education across” it is incomplete sentence.

Background:

• This introduction to Train4Health will address identified shortcomings, thereby clarifying the project’s relevance rather than presenting it as a pre-existing solution seeking validation. The core issue can be distilled into a singular problem statement: “Despite the increasing acknowledgment of behavior change support as a fundamental professional competency, there is a notable lack of understanding regarding how undergraduate students and academic educators across various disciplines and countries perceive current educational provisions and nascent educational tools.”

• In the background, emphasize the importance of stakeholder evaluation instead of merely what was created.

• Line 72: “adress this gap” should be “address this gap”

Methods:

• Study Design and Rationale: The reason for selecting focus groups instead of one-on-one interviews is not clearly mentioned. It remains uncertain if the goal was to investigate interactional dynamics (a primary reason for focus groups) or merely to collect individual viewpoints. So kindly provide a concise rationale for selecting focus groups, especially considering the online and international context.

• Research Team and Reflexivity (COREQ Item): kindly incorporate a brief reflexivity statement that recognizes the researchers’ skills and possible biases, as well as the strategies employed to address these, such as utilizing various coders and conducting consensus meetings.

• Sampling and Recruitment: kindly elucidate the reasons student leadership positions were seen as equal to senior-year status. How this could have affected viewpoints if this resulted in selection bias.

• Focus Group Composition and Conduct, Kindly include Quantity of focus groups, composition of every group, and typical duration. These are standard benchmarks according to COREQ.

• Data Collection: Microsoft Teams and Zoom are mentioned, yet distinctions in platform usage are not addressed. Kindly summarize if the choice of platform influenced participation or the quality of data in any way.

• Data Analysis: Kindly specify if the analysis was inductive, deductive, or a combination of both, how saturation was addressed, and how cross-language consistency was upheld in the analysis.

Results:

• Line 171, 22 year please rewrite to 22 years

• Clarify “other institutions in Europe (n=17)” this is vague.

• Introduce each theme with a concise analytic summary (2–3 sentences) before presenting quotations. Explicitly state how sub-themes relate to the main theme.

Discussion:

• Interpretation versus Description: The Discussion usually interprets the findings correctly, but in some instances, it reiterates results instead of synthesizing them: The initial paragraphs emphasize disparities in curricula and differences among countries without adequately transforming these observations into conceptual understandings.

• Interprofessional Education (IPE):

The lack of educator viewpoints on timing is observed but not examined.

Suggestion: Quickly explain why educators might not consider timing, such as limitations in the curriculum or the tension between institutional authority and student experience.

Strengths and Limitations Section:

• Avoid defending limitations too strongly (e.g., extensive justification of virtual methods).

Conclusions:

• Rewrite the opening sentence please to be grammatically correct.

• Some recommendations could be more explicitly tied back to participant data. Suggestion: Revise for clarity and ensure all recommendations clearly stem from the qualitative findings.

Note: Throughout this document, please ensure consistent adherence to either British or American spelling conventions (e.g., “behaviour” vs. “behavior”).

.

Reviewer #1: No

Reviewer #2: No

---

## [Author Response · Author response to Decision Letter 1]

19 Feb 2026

Response to Reviewers

“This project has received funding from the Erasmus+ Programme of the European Union under the grant agreement no. 2019–1-PT01-KA203–061389. The Funder had no role in the design of the study and collection, analysis, and interpretation of data and in writing the manuscript. The European Commission’s support for the production of this publication does not constitute an endorsement of the contents, which reflect the views only of the authors, and the Commission cannot be held responsible for any use which may be made of the information contained therein.”

“This project has received funding from the Erasmus+ Programme of the European Union under the grant agreement no. 2019–1-PT01-KA203–061389. The Funder had no role in the design of the study and collection, analysis, and interpretation of data and in writing the manuscript. The European Commission’s support for the production of this publication does not constitute an endorsement of the contents, which reflect the views only of the authors, and the Commission cannot be held responsible for any use which may be made of the information contained therein.”

We confirm that the funder statement is correct, and we have added it to the cover letter.

We have revised the manuscript accordingly by removing the “Ethical approval and consent to participate” section. The ethics information is now included only in the Methods section.

Reviewer #1

1. This study offers a valuable contribution to the field, particularly by emphasizing the important role of behavioral change input in delivering patient-centered advice—especially in cases that require interdisciplinary collaboration among various healthcare professionals.

Thank you for the positive feedback. We are pleased that the Reviewer recognises the study’s contribution, particularly the emphasis on behavioural change input to support patient-centred advice in interdisciplinary care.

2. Abstract: The abstract should be revised to improve clarity, coherence, and adherence to standard academic conventions.

We have revised the Abstract to incorporate this suggestion.

3. Text justification: Consistent text alignment is essential and should comply with the journal’s formatting guidelines.

We have revised the manuscript to ensure consistent text alignment throughout, in accordance with the journal’s formatting guidelines.

4. Use of bold text: The application of bold formatting appears excessive or unnecessary in several instances. Limiting its use will help maintain an appropriate academic tone.

We have reviewed the manuscript and limited the use of bold text, removing unnecessary instances

5. Bullet points: The format and usage of bullet points should be reviewed and revised to align with formal academic standards.

We have transformed bullet-points into running text in several sections of the paper (e.g. study objectives).

Reviewer #2

1. The abstract

1.1. Line 7: The identified research gap requires further refinement. While the existing literature addresses behavior change education and interprofessional learning, the claim that certain perspectives “remain unexplored” necessitates either modification or stronger substantiation.

We acknowledge the need for refinement in the stated research gap. We provided a three-part substantiation for the gap that we are addressing: current views on behaviour change support education, the identification of unmet needs in the curriculum, and stakeholder perspectives on the implementation of novel educational tools.

1.2. The methods section of the abstract lacks specific details concerning the structure of the focus groups, such as whether they were discipline-specific or interprofessional, and whether students and teachers were combined or separated.

We have clarified this by mentioning i) the total number of focus groups (12), ii) the fact that they were held separately for students and educators, iii) the number of groups for each stakeholder (7 and 5) and iv) the interprofessional nature of focus groups.

1.3. Line 16: Grammatical issues require attention, such as the phrase “Participants were recruited at a national and transational level,” which should be corrected to “transnational.”

We apologise for the oversight. The typo has been corrected.

1.4. Line 16: “The focus group discussions was recorded” should be “were recorded.”

We apologise for the oversight. We have corrected the sentence.

1.5. “This study highlights the unmet needs in behaviour change support education across” it is incomplete sentence.

We apologise for the oversight, which has been corrected.

2. Background

2.1. This introduction to Train4Health will address identified shortcomings, thereby clarifying the project’s relevance rather than presenting it as a pre-existing solution seeking validation. The core issue can be distilled into a singular problem statement: “Despite the increasing acknowledgment of behavior change support as a fundamental professional competency, there is a notable lack of understanding regarding how undergraduate students and academic educators across various disciplines and countries perceive current educational provisions and nascent educational tools.”

We have revised the Background to strengthen the problem-driven rationale. Specifically, we added a concise problem statement highlighting the persisting gap in understanding how undergraduate students and academic educators across disciplines and countries perceive current educational provision and emerging tools for behaviour change support (page 4, line 29-37) and repositioned the Train4Health as a response to these shortcomings.(page 5, line 38-41).

2.2. In the background, emphasize the importance of stakeholder evaluation instead of merely what was created.

We now explicitly state that developing educational products does not in itself ensure relevance of content and delivery, nor adoption (page 5, line 34-37). To support this point, we added evidence from a recent scoping review of digital methods for teaching practical skills, which emphasises that engaging students and teachers, alongside researchers, is important during development (doi: 10.1007/s44217-022-00022-x). We then use this rationale to justify our focus on exploring students’ and educators’ perspectives and contextual conditions for using the Train4Health products.

We have also changed the last paragraph of the “Background” section to support this point.

2.3. Line 72: “adress this gap” should be “address this gap”

We have corrected the sentence.

3. Methods

3.1 Study Design and Rationale: The reason for selecting focus groups instead of one-on-one interviews is not clearly mentioned. It remains uncertain if the goal was to investigate interactional dynamics (a primary reason for focus groups) or merely to collect individual viewpoints. So kindly provide a concise rationale for selecting focus groups, especially considering the online and international context.

We have clarified the rationale for choosing focus groups (page 5; lines 52-54).

3.2 Research Team and Reflexivity (COREQ Item): kindly incorporate a brief reflexivity statement that recognizes the researchers’ skills and possible biases, as well as the strategies employed to address these, such as utilizing various coders and conducting consensus meetings.

We have now expanded the Methods (lines 58-70) to provide a clearer research team and reflexivity account. We report moderators’ credentials, experience and team professional backgrounds, and we acknowledge potential influences related to pre-existing relationships, possible recognition of moderators by students, and the team’s positioning within the funded consortium. We also clarified the strategies used to address these, including standardised procedures and piloted topic guides, assistant note-taking, multiple-coder analysis using a shared codebook, and consensus meetings to agree coding decisions, themes and sub-themes.

3.3 Sampling and Recruitment: kindly elucidate the reasons student leadership positions were seen as equal to senior-year status. How this could have affected viewpoints if this resulted in selection bias.

We explained in the methods (page 6, lines 83-85) that we included student leadership roles (e.g., students’ unions/boards) as an alternative to final-year status because these students typically have substantial exposure to quality processes and cross-cohort student needs, and are often involved in discussions on educational provision. We recognise this may have introduced selection bias by over-representing students who are more engaged in governance. We have clarified this under limitations (page 27, lines 395-398).

3.4 Focus Group Composition and Conduct, Kindly include Quantity of focus groups, composition of every group, and typical duration. These are standard benchmarks according to COREQ.

In line with the COREQ checklist, we have revised the Participant selection and recruitment section to report the number of focus groups, size of each group and the duration of sessions. This information has been moved from the Results section.

3.5 Data Collection: Microsoft Teams and Zoom are mentioned, yet distinctions in platform usage are not addressed. Kindly summarize if the choice of platform influenced participation or the quality of data in any way.

We have clarified in the Data collection section that recordings were obtained via videoconferencing software depending on institutional access and availability, and that procedures were standardised across platforms (page 8; line 124). We also addressed this issue in the Discussion section, noting that we were unable to assess whether platform choice influenced participation or the quality of the data collected (page 27; line 386-391).

3.6 Data Analysis: Kindly specify if the analysis was inductive, deductive, or a combination of both, how saturation was addressed, and how cross-language consistency was upheld in the analysis.

We have clarified in the Data management and analysis section that we used a combined deductive–inductive approach to analysis (page 8; line 135). Regarding data saturation, the number of focus was determined a priori, based on our experience, the resources available and guidance from the literature. We added a paragraph in the Discussion (page 27, line 377), noting that although saturation was not employed to determine the number of focus groups, we observed increasing repetition of themes in later focus groups and that our sample (12 groups) is above the number commonly reported as sufficient for thematic saturation in these studies (often 4–8 groups). The implications of our pragmatic approach to sample size is also discussed.

4. Results

4.1 Line 171, 22 year please rewrite to 22 years

We have corrected the typo.

4.2 Clarify “other institutions in Europe (n=17)” this is vague.

We have listed the European universities reported in the sociodemographic questionnaires. We also clarify that university affiliation was not provided by 13 participants (page 9; line 152-158).

4.3 Introduce each theme with a concise analytic summary (2–3 sentences) before presenting quotations. Explicitly state how sub-themes relate to the main theme.

We have revised the Results section to introduce each theme with a summary prior to presenting illustrative quotations. We also now explicitly describe how the sub-themes relate to and structure each main theme (e.g., by clarifying the organising logic and the connections between sub-themes and the overarching theme).

5. Discussion

5.1 Interpretation versus Description: The Discussion usually interprets the findings correctly, but in some instances, it reiterates results instead of synthesizing them: The initial paragraphs emphasize disparities in curricula and differences among countries without adequately transforming these observations into conceptual understandings.

We have revised the opening of the Discussion section according to the reviewer 's suggestion.

5.2 Interprofessional Education (IPE):

The lack of educator viewpoints on timing is observed but not examined.

Suggestion: Quickly explain why educators might not consider timing, such as limitations in the curriculum or the tension between institutional authority and student experience.

We have addressed this point in the Discussion section (page 24; line 286-292)

6. Strengths and Limitations Section

6.1 Avoid defending limitations too strongly (e.g., extensive justification of virtual methods).

We have revised the Limitations section to present the constraints of virtual focus groups and remove extensive justification. We retained the citations not to downplay the limitation, but to contextualise it within the literature and support a balanced interpretation.

7. Conclusions

7.1 Rewrite the opening sentence please to be grammatically correct.

7.2 Some recommendations could be more explicitly tied back to participant data. Suggestion: Revise for clarity and ensure all recommendations clearly stem from the qualitative findings.

We have revised the Conclusions section to ensure clarity and accommodate the reviewer’s comments.

8. Note: Throughout this document, please ensure consistent adherence to either British or American spelling conventions (e.g., “behaviour” vs. “behavior”).

We apologise for the oversight and have revised the manuscript to ensure consistent use of British English spelling.

---

## [Decision Letter · Decision Letter 1]

24 Mar 2026

Behaviour change support education in chronic disease: an international focus group study with undergraduate students and academic educators in nursing, pharmacy and sport science disciplines

PONE-D-25-32206R1

Dear Dr. Félix,

We’re pleased to inform you that your manuscript has been judged scientifically suitable for publication and will be formally accepted for publication once it meets all outstanding technical requirements.

Kind regards,

Elif Ulutaş Deniz

Academic Editor

PLOS One

Additional Editor Comments (optional):

Reviewers' comments:

Reviewer's Responses to Questions

**Comments to the Author**

Reviewer #1: All comments have been addressed

Reviewer #2: All comments have been addressed

2. Is the manuscript technically sound, and do the data support the conclusions?

Reviewer #1: Yes

Reviewer #2: Yes

3. Has the statistical analysis been performed appropriately and rigorously?

Reviewer #1: I Don't Know

Reviewer #2: Yes

4. Have the authors made all data underlying the findings in their manuscript fully available?

Reviewer #1: Yes

Reviewer #2: Yes

5. Is the manuscript presented in an intelligible fashion and written in standard English?

Reviewer #1: Yes

Reviewer #2: Yes

Reviewer #1: (No Response)

Reviewer #2: (No Response)

.

Reviewer #1: No

Reviewer #2: No

---

## [Editor Report · Acceptance letter]

PONE-D-25-32206R1

PLOS One

Dear Dr. Félix,

I'm pleased to inform you that your manuscript has been deemed suitable for publication in PLOS One. Congratulations! Your manuscript is now being handed over to our production team.

Kind regards,

on behalf of

Dr. Elif Ulutaş Deniz

Academic Editor

PLOS One